# Biochar Application Increases Labile Carbon and Inorganic Nitrogen Supply in a Continuous Monocropping Soil

**Rong Huang** [1,2], **Bing Li** [1,*], **Yulan Chen** [3], **Qi Tao** [1], **Qiang Xu** [1], **Denghong Wen** [1,4], **Xuesong Gao** [1], **Qiquan Li** [1], **Xiaoyan Tang** [1] **and Changquan Wang** [1]

1   College of Resources, Sichuan Agricultural University, Chengdu 611130, China; 14624@sicau.edu.cn (R.H.); 14449@sicau.edu.cn (Q.T.); 14434@sicau.edu.cn (Q.X.); jhcwk99@163.com (D.W.); xuesonggao@sicau.edu.cn (X.G.); liqq@lreis.ac.cn (Q.L.); xytang@sicau.edu.cn (X.T.); wcquan@sicau.edu.cn (C.W.)
2   Chongqing Key Laboratory of Soil Multi-Scale Interfacial Process, Chongqing 400715, China
3   Liangshan Branch of Sichuan Tobacco Corporation, Xichang 615000, China; 2020106002@stu.sicau.edu.cn
4   Liupanshui Bureau of Agriculture and Rural Areas, Liupanshui 553000, China
*   Correspondence: benglee@sicau.edu.cn

**Abstract:** Biochar is an effective method for increasing soil carbon (C) sequestration and nitrogen (N) supply under continuous monocropping. To investigate the impact of biochar placement methods on soil C and N, a one-year field experiment with five treatments was conducted including control, mineral fertilizers only (F), biochar hole placement (BFH; biochar applied to the soil layer at 5–10 cm) + F, biochar band placement (BFB; biochar applied to the soil layer at 15–20 cm) + F, and biochar band and hole placement + F (BFBH). The results showed that, regardless of the placement method, biochar application increased soil total organic C (TOC) and C pool management index by 6.9–39.7% and 4.1–36.1%, respectively, especially for dissolved organic C (DOC; 6.9–51.3%), readily oxidizable C (ROC; 2.4–46.4%), and microbial biomass C (MBC; 10.4–41.7%). Single biochar placement methods significantly influenced DOC, MBC, and ROC contents of both soil layers in the rank order of BFH ≈ BFBH > BFB at 0–15 cm and BFB ≈ BFBH > BFH at 15–30 cm. Soil TN and microbial biomass N (MBN) mainly accumulated at the site of biochar placement. The increased soil TOC:TN and MBC:N ratios under biochar treatments promoted inorganic N immobilization and reduced the loss of ammonium N and nitrate N ($NO_3^-$-N) through leaching at the early stage of tobacco growth. Biochar-adsorbed N was remobilized at a later period (vigorous growth stage and maturity), possibly causing the slow decrease in $NO_3^-$-N content. Additionally, soil C and N pools were significantly influenced by the main effects of soil layer and growth stage. Overall, biochar application increased soil C and N pools and inorganic N supply through N remobilization. However, the increased labile organic C content and microbial activity may prevent C sequestration in biochar-amended soils.

**Keywords:** biochar; soil labile organic C; soil inorganic N; placement method

## 1. Introduction

Biochar is a carbonaceous material that is widely used for soil amendment [1,2] and has gained attention from the scientific community and policy makers [1,3]. Due to the high surface area, cation exchange capacity, and N/P contents of biochar [4], there is ample evidence that biochar application alters the physicochemical and biological properties of soil [4], thereby influencing the sequestration of soil organic carbon (SOC), soil quality, emission of greenhouse gases, bioavailability of heavy metal pollutants, and the diversity and activity of soil microorganisms [1,2,5–10]. The large input of biochar-derived C into soil influences the turnover of C and nitrogen (N) pools, especially over the short term [11]. Research has documented that the improvement of C sequestration by biochar application is even higher than that by straw application [12]. Biochar was also shown to exert a positive priming effect on soil organic matter (SOM) mineralization within a short time

(2 months) [1]. As a component of bulk SOC, soil labile organic C (LOC) is sensitive to soil management practices and is used as an index of soil quality and productivity [3,13]. The soil C pool management index (CPMI) is also used as a measure of the efficacy of soil management practices in promoting soil quality [3,14], which is dependent on SOC content and C lability. In addition to the soil C pool, biochar application also impacts N cycling in soils because of the close relationships between N availability, N turnover rate, and SOC [11,15,16]. For example, biochar application was found to enhance the denitrification rate [17], increase the N uptake by plants [10], decrease dissolved organic N losses [8,18,19], and cause net N immobilization in soils [16]. Meanwhile, inorganic N could be captured by biochar via adsorption (either physisorption or chemisorption) [20].

Continuous monocropping leads to soil degradation, which reduces the soil aptitude for production [9,21]. Tobacco (*Nicotiana tabacum* L.) is vulnerable to growth obstacles under continuous cultivation [21–23]. China is the world's largest producer of tobacco; planting this crop is especially important for the local economy in many poverty-stricken areas of central and western China [24]. Continuous cropping obstacles, such as soil nutrient reduction and decreased SOM, yield, and product quality, have caused the wilting and death of plants, thus jeopardizing the agricultural sustainability of this crop [25,26]. Soil amendment (e.g., with biochar) can help to overcome these obstacles.

While many studies have investigated the effects of the type and dosage of mineral N fertilizer on crop productivity [27–29], the placement method of fertilizers has been largely overlooked, especially the role of different placement methods of biochar. Recent studies have demonstrated that different methods of biochar placement, such as uniform mixing in topsoil or subsoil and deep banding into the rhizosphere, influence soil hydraulic properties (e.g., soil water retention and hydraulic conductivity) and soil structure [30,31]. This could in turn affect the transport of biochar and alter the partitioning of nutrients (e.g., C and N) between soil layers. For example, biochar mixed into 10–20 cm of subsurface soil reduced nitrate leaching [30]. However, the effects of biochar placement on soil C and N remain unclear, although appropriate placement may potentially optimize soil C and N contents in continuously cropped fields [32].

We hypothesized that the placement method of biochar in soil influences the concentrations of soil C and N fractions. The aims of this study were as follows: (1) to examine the responses of soil C and N fractions to short-term application of biochar in a continuously cropped field; and (2) to compare the effects of different biochar placement methods on soil LOC and N.

## 2. Materials and Methods

### 2.1. Study Site and Soil Characteristics

The experiment was conducted in Ganhai Township, Yanyuan County, Liangshan Prefecture, Sichuan Province, China (101°29′35.583″ E, 27°29′48.107″ N). This site, which is at 2349 m above sea level, has a subtropical monsoon climate with a mean annual temperature of 13.2 °C and a mean precipitation of 727.3 mm. The weather data are shown in Figure 1. The soil used as the arable layer in a local tobacco field was Typic Dystrochrept. The chemical properties of the soil at the study site were as follows: pH of $6.38 \pm 0.54$ (soil:water = 1:1), SOM of $22.46 \pm 3.21$ g kg$^{-1}$, soil alkali-hydrolyzable N of $87.12 \pm 9.54$ mg kg$^{-1}$, soil available phosphorus (P) of $9.48 \pm 1.02$ mg kg$^{-1}$, and soil available potassium (K) of $84.15 \pm 6.72$ mg kg$^{-1}$.

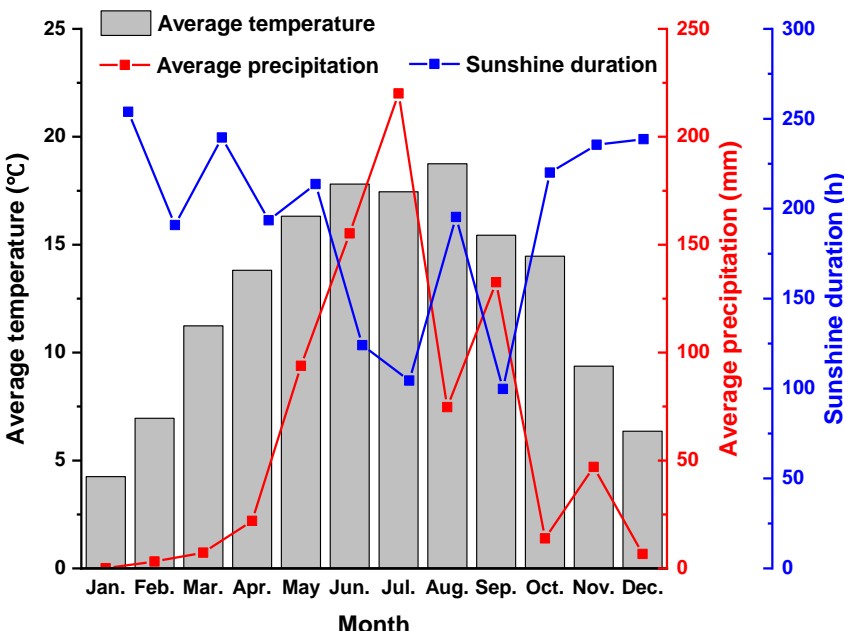

**Figure 1.** Average temperature, average precipitation, and sunshine duration at study site.

### 2.2. Experimental Materials

Biochar was obtained from Sichuan Meijia Biomass Energy Co., LTD, China, and was prepared by colza straw pyrolysis at a temperature of 400–500 °C under anaerobic conditions. The N, P, and K contents of biochar were 5.9 g kg$^{-1}$ (total N (TN)), 0.91 g kg$^{-1}$ (total P), and 26.0 g kg$^{-1}$ (total K), respectively; and the C and ash contents were 481.4 and 20.8 g kg$^{-1}$, respectively. The pH of biochar was 9.65 (measured at biochar:water = 1:10). The flue-cured tobacco (*Nicotiana tabacum* L.) variety used in this study was KRK26.

The mineral fertilizers used in this study were a compound fertilizer for tobacco (N:P$_2$O$_5$:K$_2$O = 9:9:27), kalium nitrate (14% N, 44% K$_2$O), and potassium sulfate (51% K$_2$O), which were purchased from Sichuan Jinye Fertilizer Co. (Chengdu, China).

### 2.3. Experimental Design

The five treatments, which were arranged in a completely randomized design with three replicates (15 plots in total), were as follows: (1) control (no biochar or mineral fertilizers); (2) only mineral fertilizers at the recommended dosage (F); (3) 7600 kg hm$^{-2}$ biochar band placement + F (BFB; biochar applied to soil layer at 15–20 cm); (4) 7600 kg hm$^{-2}$ biochar hole placement + F (BFH; biochar applied to soil layer at 5–10 cm); and (5) 3800 kg hm$^{-2}$ biochar band placement and 3800 kg hm$^{-2}$ biochar hole placement + F (BFBH). The same amount of mineral fertilizer was applied in a circle around each tobacco plant (4–5 cm radius) in all treatments (except for the control) (Table 1). The amounts were based on local fertilization standards and were as follows: special compound fertilizer for tobacco at 675 kg hm$^{-2}$, kalium nitrate fertilizer at 225 kg hm$^{-2}$, and potassium sulfate fertilizer at 225 kg hm$^{-2}$. Detailed information on fertilization is shown in Table 1.

A total of 150 flue-cured tobacco plants were transplanted in the ridges (30 cm) of each plot (20 m long by 4.8 m wide) with a line spacing of 120 cm and row spacing of 55 cm (Figure 2) (2250 flue-cured tobacco plants were planted in total). In the BFB treatment, a band was created under the soil ridge at a depth of 15–20 cm, and biochar (7600 kg hm$^{-2}$) was applied in the band at one time. In the BFH treatment, a hole was created at a depth of 5–10 cm from the soil ridge, and the biochar (7600 kg hm$^{-2}$) was applied in the hole at one time. In the BFBH treatment, the band and hole were prepared as described above, and 3800 kg hm$^{-2}$ biochar was applied. The biochar used in the study was applied as the basal fertilizer before transplanting the tobacco plants (in April 2017).

**Table 1.** Detailed information on fertilization under different treatments.

| Treatment | Basal Fertilizer (kg ha$^{-1}$) | | Topdressing (kg ha$^{-1}$) | | |
| :---: | :---: | :---: | :---: | :---: | :---: |
| | Biochar | Compound Fertilizer | Compound Fertilizer | Kalium Nitrate Fertilizer | Potassium Sulfate Fertilizer |
| Control | 0 | 0 | 0 | 0 | 0 |
| F | 0 | 405 | 270 | 225 | 225 |
| BFB | 7600 | 405 | 270 | 225 | 225 |
| BFH | 7600 | 405 | 270 | 225 | 225 |
| BFBH | 7600 | 405 | 270 | 225 | 225 |

Control, no biochar or mineral fertilizers; F, only mineral fertilizers at the recommended dosage; BFB, biochar band placement + F; BFH, biochar hole placement + F; BFBH, BFB + BFH + F.

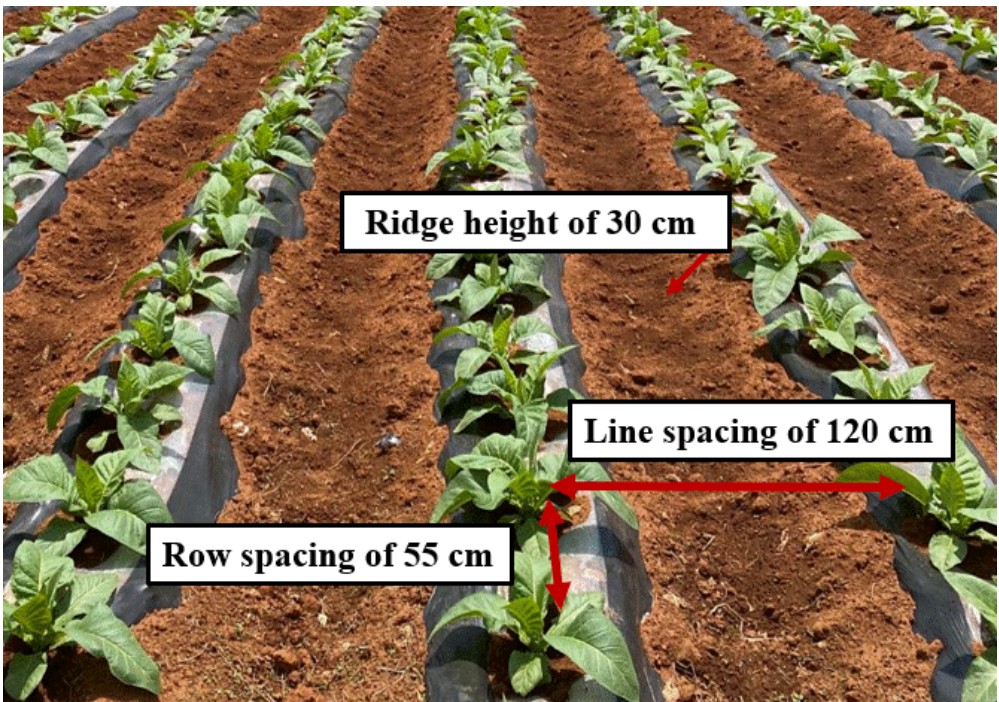

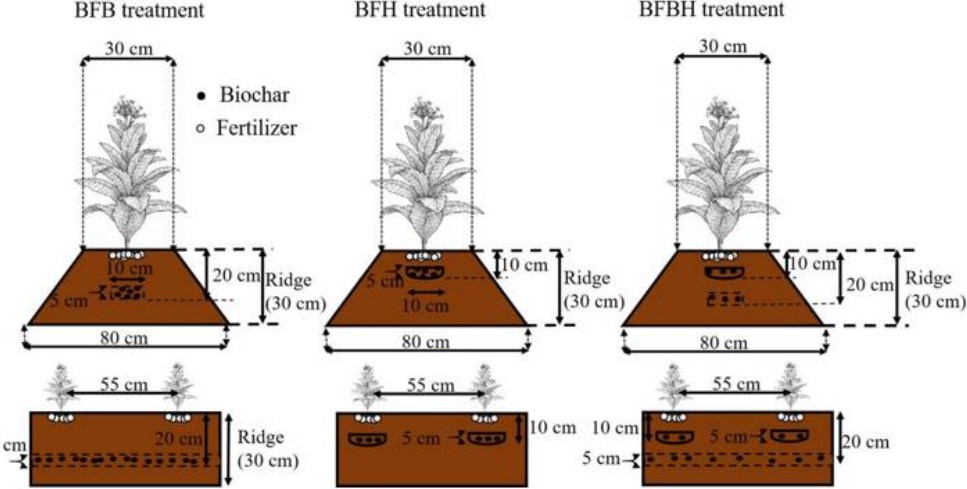

**Figure 2.** Experimental design.

### 2.4. Soil Sampling and Measurements

Soil samples were collected at a distance of 10–15 cm from each tobacco plant after the removal of organic debris. Two soil layers, including topsoil (0–15 cm) and subsoil (15–30 cm), were sampled at rosette, vigorous, and squaring growth stages, and at plant maturity (total number of raw samples was 2 layers × 5 samples per plot × 15 plots × 4 stages = 600). Five random samples from each plot and each layer were pooled for the analysis of soil LOC fractions (microbial biomass C (MBC) [33], dissolved organic C (DOC) [6,7], readily oxidizable C (ROC) [6,7], and particle organic C (POC) [34]) and soil mineral N fractions (ammonium N ($NH_4^+$-N) and nitrate N ($NO_3^-$-N)) (120 pooled samples were measured). The fresh soil samples were stored at 4 °C. Air-dried soil was passed through 1- and 0.25-mm sieves and used for the analysis of total soil organic C (TOC), soil TN, and POC.

Soil TOC content was measured by $K_2Cr_2O_7$ oxidation and $FeSO_4$ titration [35]. Soil TN content was determined by micro-Kjeldahl digestion using a N analyzer automatic distillation device (KDN-102C; Shanghai QianJian Instruments Co., Shanghai, China). Soil MBC and MBN contents were determined by $K_2SO_4$ extraction with the chloroform fumigation–extraction protocol [33]. Soil POC was determined by particle size fractionation [34]. Soil DOC, ROC, $NH_4^+$-N, and $NO_3^-$-N measurements have been described in our previous studies [6,7].

### 2.5. Calculations and Statistical Analyses

The soil C pool management index (CPMI) is a measure of SOC variation in response to soil management practices [13]. CPMI was calculated according to Yang et al. (2018) [3], with the reference replaced by the control in the present study:

$$CMPI = CPI \times LI \times 100 \tag{1}$$

$$CPI = TOC_{treatment}/TOC_{control} \tag{2}$$

$$LI = L_{treatment}/L_{control} \tag{3}$$

$$L = ROC/NLC \tag{4}$$

$$NLC = TOC - ROC \tag{5}$$

where, TOC and ROC were measured in g kg$^{-1}$; NLC is the nonlabile C content (g kg$^{-1}$); L is C lability; $L_{treatment}$ and $L_{control}$ are the C lability of the sample and reference soils, respectively; LI is the lability index (ratio of C lability of the sample soil to that of the reference soil); CPI is the C pool index; and TOC treatment and TOC control are the TOC contents of the sample and reference soils, respectively.

All data were analyzed using SPSS v18.0 software (SPSS Inc., Chicago, IL, USA). The normality of datasets was assessed by the Shapiro–Wilk test. Differences among treatment groups were evaluated by 1-way analysis of variance (ANOVA) combined with the least significant difference test ($p < 0.05$). The effects of treatment, soil layer, and growth stage and their interaction on soil properties were assessed by 3-way ANOVA using SPSS v23.0 software.

## 3. Results

### 3.1. Soil TOC and LOC Fractions

The application of biochar (BFB, BFH, and BFBH) increased soil TOC content compared to the control and F treatments (Figure 3a). In the soil layer at 0–15 cm, TOC content was significantly lower under the BFB treatment (13.10–16.66 g·kg$^{-1}$) than under the BFH and BFBH treatments, except in the vigorous stage of tobacco plant growth ($p < 0.05$). However, there was no significant difference between BFH and BFBH treatments except at the rosette stage. A lower TOC content was found in the soil layer at 15–30 cm compared to the layer at 0–15 cm ($p < 0.05$). The highest TOC content at 15–30 cm was found under the BFB

treatment, followed by the BFBH treatment. The main effects of treatment, soil layer, and growth stage on soil TOC, as well as the interaction effect of these three factors, were all significant ($p < 0.05$; Table 2).

Biochar application significantly increased soil DOC content relative to the control and F treatments regardless of the soil layer (0–15 or 15–30 cm) ($p < 0.05$; Figure 3b). At 0–15 cm, the highest soil DOC content was found under the BFH treatment and was 33.3–51.3% higher than that under the F treatment ($p < 0.05$). However, at 15–30 cm, soil DOC content under the BFH treatment (45.09–56.46 mg kg$^{-1}$) was lower than that under the BFB and BFBH treatments. The highest soil MBC content at 0–15 cm was found under the BFH treatment (Figure 3c), and soil MBC was significantly higher under the BFH and BFBH treatments than under the BFB treatment ($p < 0.05$). However, at 15–30 cm, soil MBC was lower under the BFH treatment than under the BFB and BFBH treatments ($p < 0.05$). At 0–15 cm, there was no significant difference between F and BFB treatments except at the rosette stage of tobacco plant growth, but soil ROC content was significantly higher under the BFH and BFBH treatments than under the BFB treatment ($p < 0.05$; Figure 3d). The highest soil ROC content at 15–30 cm was under the BFB treatment, followed by the BFBH treatment. Compared to the F treatment, at 0–15 cm, POC content was higher by 0.23–0.49 g kg$^{-1}$ under the BFB and BFBH treatments, and by 0.82–1.01 g kg$^{-1}$ under the BFH treatment ($p < 0.05$). Soil POC content at 15–30 cm was higher under the BFB treatment (Figure 3e). The results of the three-way ANOVA showed that soil DOC and ROC contents were influenced by the interaction effect of treatment, soil layer, and growth stage ($p < 0.05$; Table 2).

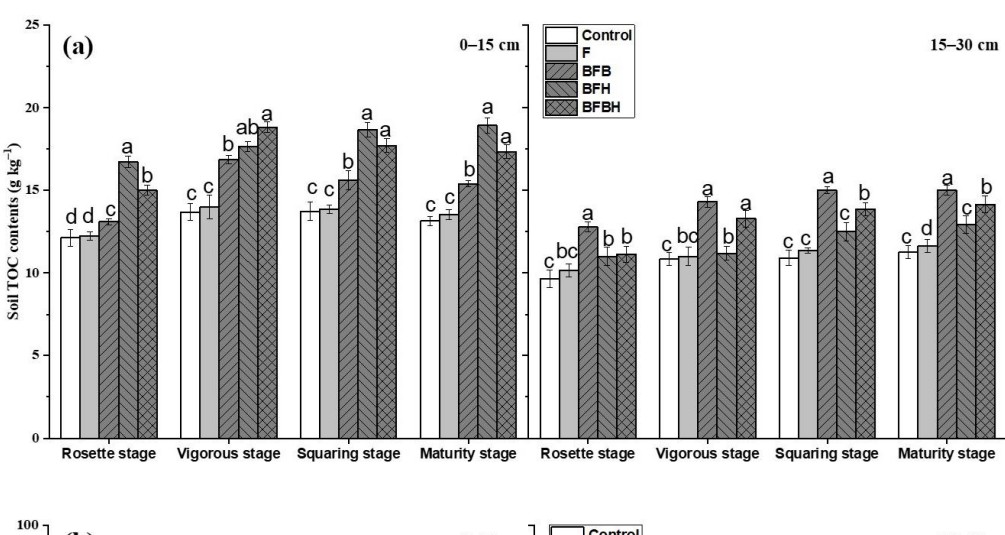

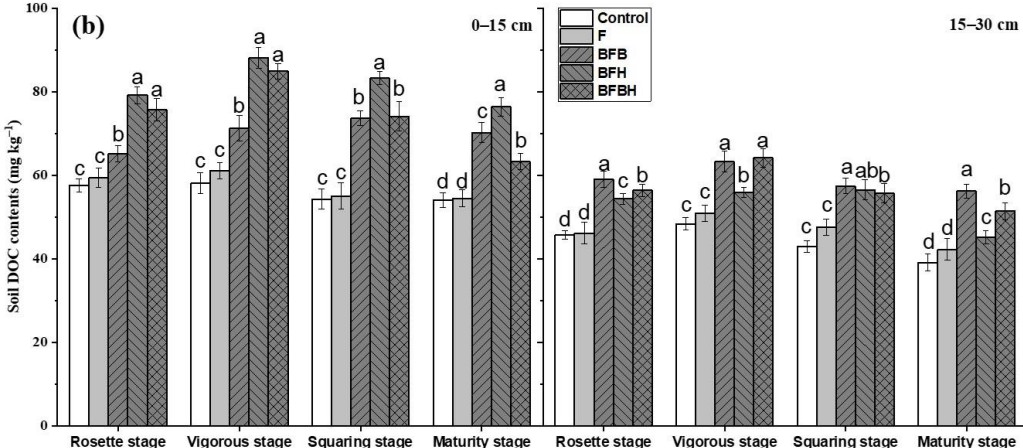

**Figure 3.** *Cont.*

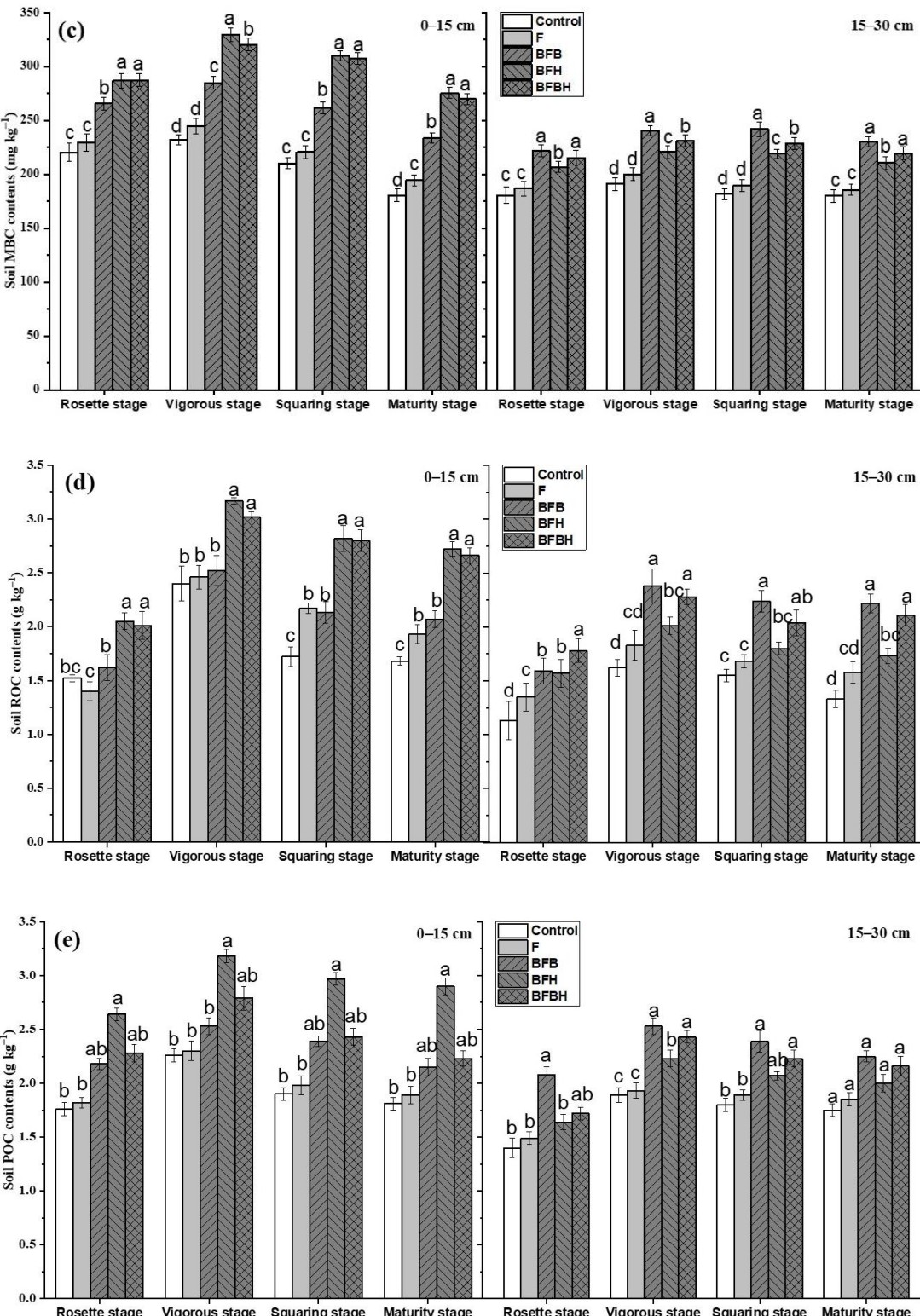

**Figure 3.** Soil TOC, DOC, MBC, ROC, and POC under different treatments. (**a–e**): Soil TOC (**a**), DOC (**b**), MBC (**c**), ROC (**d**), and POC (**e**) in soil layers at 0–15 and 15–30 cm were measured. Different lowercase letters represent significant differences among treatments at the same soil layer and growth stage ($p < 0.05$). Control, no biochar or mineral fertilizers; F, only mineral fertilizers at the recommended dosage; BFB, biochar band placement + F; BFH, biochar hole placement + F; BFBH, BFB + BFH + F.

**Table 2.** Results of 3-way ANOVA (*p* values) testing the effects of treatment, soil layer, growth stage, and their interaction on soil properties.

| | TOC | DOC | MBC | ROC | POC | TN | $NO_3^--N$ | $NH_4^+-N$ | MBN | TOC:TN | MBC:N |
|---|---|---|---|---|---|---|---|---|---|---|---|
| Treatment (T) | 16,287 * | 191.8 * | 684.1 * | 113.7 * | 17.6 * | 78.0 * | 580.5 * | 850.3 * | 83.7 * | 115.2 * | 61.5 * |
| Soil layer (S) | 7108 * | 53.0 * | 171.3 * | 157.9 * | 13.4 * | 51.8 * | 108.7 * | 291.5 * | 94.0 * | 300.1 * | 82.4 * |
| Growth stage (G) | 82,789 * | 921.5 * | 2387.9 * | 314.7 * | 36.2 * | 597.6 * | 73.5 * | 157.3 * | 733.0 * | 359.0 * | 68.3 * |
| T × S | 350 * | 4.4 * | 3.3 * | 1.5 | 0.2 | 0.9 | 10.0 * | 21.5 * | 0.3 | 6.4 * | 3.8 * |
| T × G | 6129 * | 40.7 * | 169.4 * | 36.2 * | 7.5 * | 8.8 * | 152.7 * | 161.5 * | 29.7 * | 65.2 * | 42.8 * |
| S × G | 754 * | 0.5 | 67.9 * | 14.4 * | 1.0 | 18.6 * | 3.0 * | 4.5 * | 10.9 * | 16.0 * | 47.1 * |
| T × S × G | 79 * | 2.6 * | 1.6 | 2.5 * | 0.1 | 1.7 | 1.6 | 3.2 * | 0.8 | 2.7 * | 3.6 * |

Data represent the F statistic for the results of the 3-way ANOVA; the asterisk (*) indicates a significant effect at the level of *p* < 0.05.

### 3.2. Soil CPMI

Biochar application significantly affected soil NLC content and CPMI (Table 3). In the soil layer at 0–15 cm, the highest NLC content and CPMI were observed under the BFH treatment; these were 33.9% and 36.1% higher, respectively, than under the F treatment (*p* < 0.05); however, at 15–30 cm, the values were 29.7% and 30.7% higher, respectively (*p* < 0.05). There was no significant difference in NLC content or CPMI between F and BFH treatments at 15–30 cm.

**Table 3.** Soil carbon pool management index under different treatments.

| Soil Layer | Treatment [†] | NLC (g kg$^{-1}$) | L | LI | CPI | CPMI |
|---|---|---|---|---|---|---|
| 0–15 cm | Control | 11.35Ac | 0.16Aa | | | |
| | F | 11.43Ac | 0.17Aa | 1.08Aa | 1.02Ac | 110.30Ab |
| | BFB | 13.17Ab | 0.16Aa | 0.99Aa | 1.16Bb | 114.77Bb |
| | BFH | 15.31Aa | 0.18Aa | 1.10Aa | 1.37Ba | 150.12Aa |
| | BFBH | 14.61Aa | 0.18Aa | 1.13Aa | 1.31Aa | 147.70Aa |
| 15–30 cm | Control | 9.26Bc | 0.15Ab | | | |
| | F | 9.44Bc | 0.17Aab | 1.13Aa | 1.04Ad | 116.82Ab |
| | BFB | 12.18Aa | 0.17Aab | 1.14Aa | 1.34Aa | 152.64Aa |
| | BFH | 10.14Bbc | 0.18Aab | 1.17Aa | 1.12Ac | 130.37Ab |
| | BFBH | 11.07Bab | 0.19Aa | 1.24Aa | 1.23Ad | 152.21Aa |

Different lowercase letters within the same column indicate significant differences among different treatments at the same soil layer (*p* < 0.05); different uppercase letters represent significant differences among different soil layers under the same treatment and growth stage (*p* < 0.05). [†] Control, no biochar or mineral fertilizers; F, only mineral fertilizers at the recommended dosage; BFB, biochar band placement + F; BFH, biochar hole placement + F; BFBH, BFB + BFH + F. Abbreviations: CPI, carbon pool index; CPMI, carbon pool management index; L, soil carbon lability; LI, lability index (ratio of soil sample carbon lability to reference soil carbon lability); NLC, nonlabile carbon.

### 3.3. Soil N Fractions

Fertilization (F, BFB, BFH, and BFBH)—especially the BFH treatment—increased soil TN content in the soil layer at 0–15 cm (Figure 4a). At 15–30 cm, TN content was 3.5–8.0% higher under the BFB treatment than under the F treatment (*p* < 0.05). Compared to the F treatment, $NO_3^-$-N content at 0–15 cm was reduced by 15.5–43.6% under the BFB and BFBH treatments, respectively (*p* < 0.05; Figure 4b). Besides the control, the lowest soil $NO_3^-$-N content was observed under the BFH treatment at 15–30 cm. $NH_4^+$-N at 0–15 cm was significantly lower under the BFB and BFBH treatments than under the F and BFH treatments (*p* < 0.05; Figure 4c); however, at 15–30 cm, a lower $NH_4^+$-N content was observed under the BFH and BFBH treatments than under the F and BFB treatments (*p* < 0.05). Compared to the F treatment, biochar application increased soil MBN content at 0–15 cm and, to a lesser extent, at 15–30 cm (Figure 4d). The highest soil MBN content was observed under the BFH treatment in both layers.

*3.4. Ratio of Soil C to N*

The TOC:TN ratio, which is a measure of soil organic quality and nutrient limitations [36,37], was higher under biochar (BFB, BFH, and BFBH) treatments compared to the F treatment in the soil layer at 0–15 cm; the difference was especially significant under the BFH treatment ($p < 0.05$; Table 4). At 15–30 cm, the TOC:TN ratio increased with the growth of the tobacco plant and was highest under the BFB treatment, followed by the BFBH treatment ($p < 0.05$).

**Table 4.** Soil TOC:TN and MBC:N ratios in different soil layers under different treatments at each growth stage.

| Treatment [†] at Different Soil Layers | TOC:TN | | | | MBC:N | | | |
|---|---|---|---|---|---|---|---|---|
| | Rosette Stage | Vigorous Stage | Squaring Stage | Maturity | Rosette Stage | Vigorous Stage | Squaring Stage | Maturity |
| 0–15 cm | | | | | | | | |
| Control | 11.89Ac | 14.27Ac | 14.61Ac | 15.13Ac | 3.31Ba | 3.28Bb | 3.35Ac | 3.04Bd |
| F | 11.25Ac | 13.20Ad | 14.61Ac | 15.22Ac | 3.06Bb | 3.14Bc | 3.18Ad | 3.03Bd |
| BFB | 13.23Ab | 15.49Ab | 15.32Ab | 15.11Bc | 3.43Aa | 3.49Aab | 3.58Abc | 3.48Bc |
| BFH | 14.42Aa | 15.63Ab | 17.78Aa | 18.20Aa | 3.25Bab | 3.55Aa | 3.66Ab | 3.85Ab |
| BFBH | 13.54Ab | 17.28Aa | 17.19Aa | 17.00Ab | 3.52Aa | 3.73Aa | 3.91Aa | 3.96Aa |
| 15–30 cm | | | | | | | | |
| Control | 11.34Bb | 12.63Bc | 13.14Bc | 13.58Bc | 3.52Aa | 3.45Ab | 3.30Ab | 3.98Aa |
| F | 11.29Ab | 12.10Bc | 12.91Bc | 13.39Bc | 3.49Aa | 3.48Ab | 3.32Ab | 3.99Aa |
| BFB | 13.32Aa | 14.61Ba | 15.81Aa | 16.68Aa | 3.19Bb | 3.41Ab | 3.34Bb | 3.92Aa |
| BFH | 11.97Bb | 12.44Bc | 13.44Bc | 15.06Bb | 3.53Aa | 3.57Aa | 3.62Aa | 4.04Aa |
| BFBH | 11.99Bb | 13.99Bb | 14.92Bb | 15.91Ab | 3.50Aa | 3.60Aa | 3.56Ba | 4.02Aa |

Different lowercase letters within the same column indicate significant differences among different treatments at the same soil layer and growth stage ($p < 0.05$); different uppercase letters represent significant differences among different soil layers under the same treatment at the same growth stage ($p < 0.05$). [†] Control, no biochar or mineral fertilizers; F, only mineral fertilizers at the recommended dosage; BFB, biochar band placement + F; BFH, biochar hole placement + F; BFBH, BFB + BFH + F.

Biochar application increased the MBC:N ratio in the soil layer at 0–15 cm relative to the F treatment, with a higher ratio under the BFBH treatment than under the BFB and BFH treatments (Table 4). At 15–30 cm, the MBC:N ratio was reduced by 0.07–0.30 under the BFB treatment relative to the F treatment. Meanwhile, the ratio was higher under the BFH treatment than under other treatments, especially at the vigorous and squaring stages of tobacco plant growth. There were main and interaction effects of treatment, soil layer, and growth stage on TOC:TN and MBC:N ratios ($p < 0.05$; Table 2).

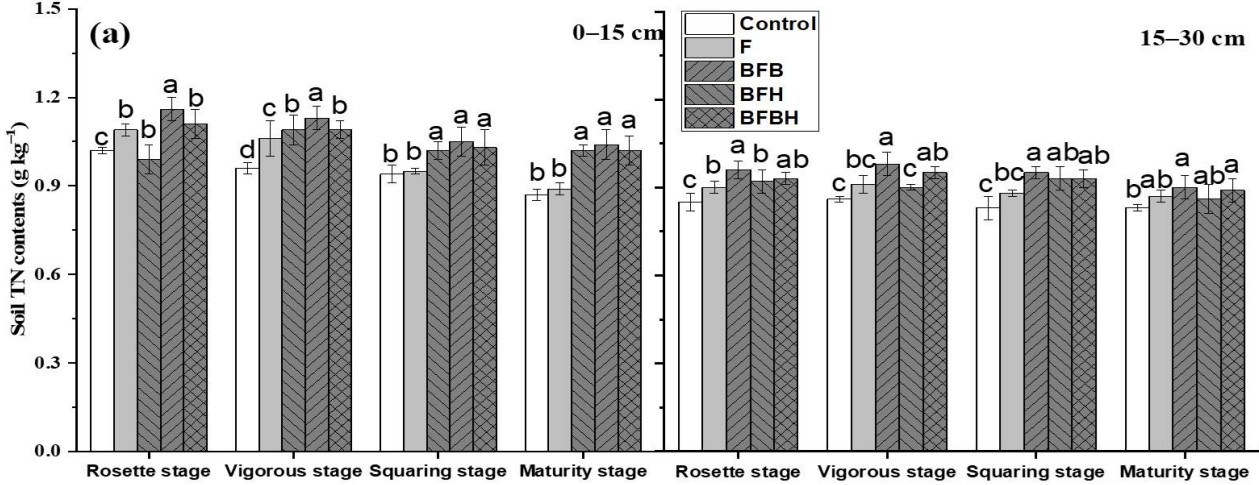

**Figure 4.** *Cont.*

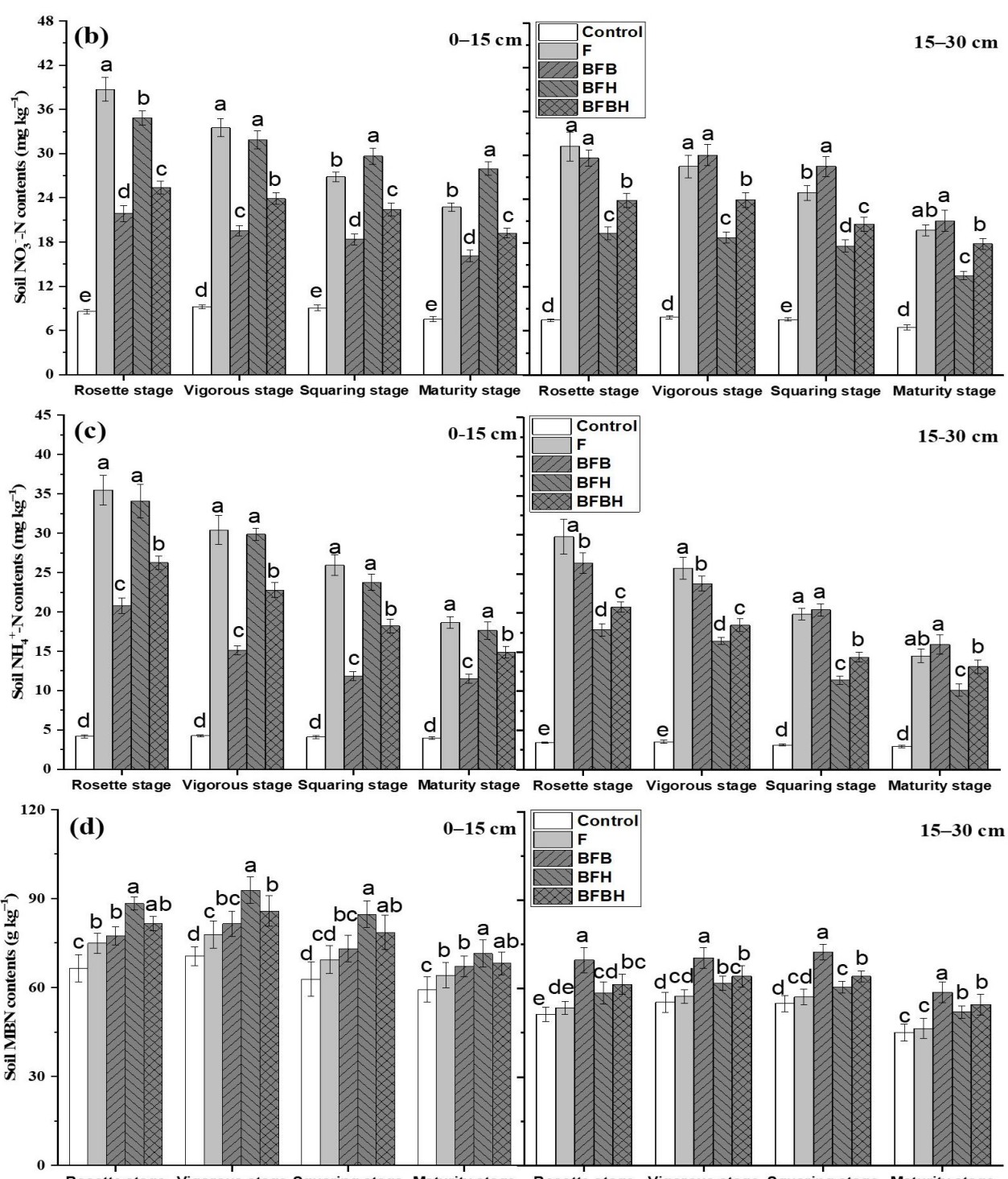

**Figure 4.** Soil TN, $NO_3^-$-N, $NH_4^+$-N, and MBN contents under different treatments. (**a**–**d**): Soil TN (**a**), $NO_3^-$-N (**b**), $NH_4^+$-N (**c**), and MBN (**d**) contents in soil layers at 0–15 and 15–30 cm were measured. Different lowercase letters represent significant differences among treatments at the same soil layer and growth stage ($p < 0.05$). Control, no biochar or mineral fertilizers; F, only mineral fertilizers at the recommended dosage; BFB, biochar band placement + F; BFH, biochar hole placement + F; BFBH, BFB + BFH + F.

*3.5. Crop Yield*

Compared with the F treatment, biochar application increased the crop yield by 4.8–12.4%, especially for BFH and BFBH (Figure 5). The highest crop yield was observed in

the BFBH treatment with the value of 2250.5 kg ha$^{-1}$, which was significantly higher than other treatments, except for the BFH treatment ($p < 0.05$).

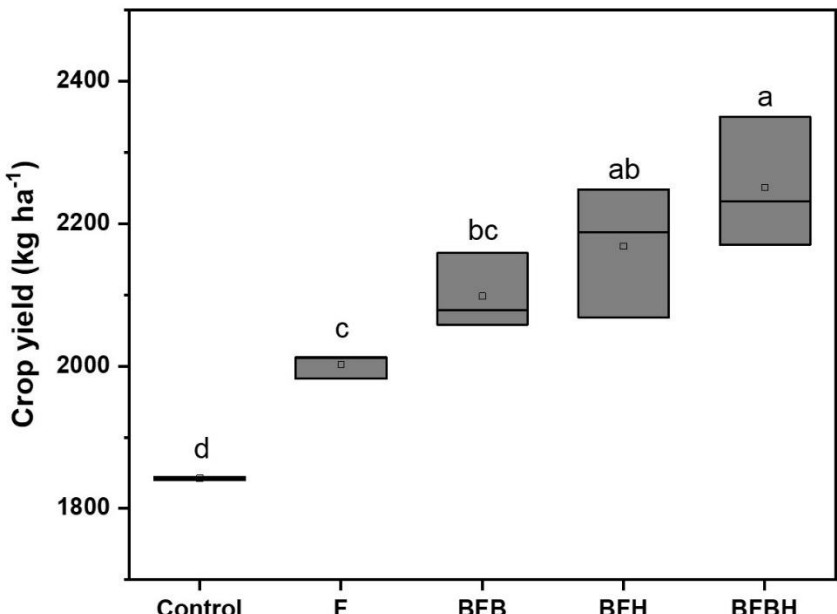

**Figure 5.** Crop yield under different treatments. Different lowercase letters represent significant differences among treatments ($p < 0.05$). Control, no biochar or mineral fertilizers; F, only mineral fertilizers at the recommended dosage; BFB, biochar band placement + F; BFH, biochar hole placement + F; BFBH, BFB + BFH + F.

## 4. Discussion

### 4.1. Effect of Biochar on the SOC Pool in a Tobacco Field

In the present study, biochar application increased SOC content in a continuously cropped tobacco field. Compared to the no-biochar (F) condition, TOC, DOC, MBC, ROC, and POC contents and CPMI were increased by biochar application at the 0–15 and 15–30 cm soil layers (Table 3), which is consistent with previous reports [38–40]. The incorporation of biochar into soil can cause short-term changes in the turnover of SOC fractions [41–43] and suppress labile organic matter mineralization through physical stabilization (e.g., sorption), thereby increasing the SOC pool [43,44]. Additionally, the high content of organic C in biochar (481.4 g·kg$^{-1}$)—especially aromatic organic compounds that are not readily biodegraded [3]—is evidenced by the higher NLC content of soil observed after biochar application in this study (Table 3). TOC content and CPMI were also increased in soil treated with biochar (BFB, BFH, and BFBH). POC is an energy source for microorganisms and a reservoir of relatively labile SOC and plant nutrients [3,45]. Biochar can remain in the form of soil POC for years, or even decades or centuries, because of the recalcitrance of biochar [46] and the enhancement of soil aggregation by its application [47]. Soil POC content was indeed higher under biochar treatments relative to the control, although the effect was not statistically significant, implying that short-term application is not sufficient for biochar to persist in soil in the form of POC.

Different fertilizer placement methods may influence the efficiency of nutrient utilization [28]. In general, placing fertilizer into the soil (e.g., through band or hole application under topsoil) increases its uptake by plants, reduces nutrient loss, and improves the competitive advantage of the crop over weeds compared to a broadcast application [28,48]. In the present study, different biochar placement methods altered the distribution of SOC fractions in the soil layers at 0–15 and 15–30 cm (Figure 3). Biochar applied by hole placement mainly caused SOC to accumulate in the layer at 0–15 cm, whereas band placement caused accumulation at 15–30 cm. This suggests that SOC does not readily mobilize along the

soil profile but accumulates near the site of biochar placement (i.e., the soil layer to which the biochar was added) in this study. The potential reasons are the limited precipitation or irrigation events [28] and the short-term application during the tobacco growth period (Figure 1). In this study, soil ROC content at 0–15 cm (or at 15–30 cm) did not significantly differ under the BFBH treatment compared to the BFH or BFB treatments (Figure 3d), implying that the placement method did not greatly influence soil ROC regardless of the amount of biochar that was applied into the hole (3800 kg hm$^{-2}$ in the BFBH treatment and 7600 kg hm$^{-2}$ in the BFH treatment) or band (3800 kg hm$^{-2}$ in the BFBH treatment and 7600 kg hm$^{-2}$ in the BFB treatment). Similarly, the biochar placement method did not significantly affect soil POC, as evidenced by the nonsignificant differences among BFB, BFH, and BFBH treatments (Figure 3e). This may be attributable to the fact that the short-term application of biochar was not sufficient to alter soil POC concentration.

In contrast, different single biochar placement methods (i.e., BFB and BFH treatments) significantly influenced the DOC, MBC, and ROC contents of the soil layers at 0–15 and 15–30 cm. This may be the result of water retention in biochar-supplemented soil, as biochar application has been shown to increase soil water retention at the boundary layers of soil treated with biochar (5–10 cm in the BFH treatment and 15–20 cm in the BFB treatment) because of the hydrophobicity of constituent organic molecules [49]. Thus, more C was released, and microorganism's activity was stimulated in an impoverished and dry soil, resulting in higher soil DOC and MBC contents following biochar application. Additionally, the biochar used in this study was pyrolyzed at 400–500 °C with an anaerobic condition, which resulted in an incomplete oxidization [3]. Accordingly, ROC fractions released from biochar itself could contribute to increases in soil ROC. Overall, biochar application increased the C pool but mainly induced the accumulation of labile C in the supplemented soil layer, indicating that short-term application may not promote the sequestration of stable C, regardless of the biochar placement method.

Soil SOC fractions varied over the growth period of the tobacco, with the lowest content at the rosette stage (Figure 3). This may be explained by the priming effects of biochar application, whereby native SOC or labile biochar compounds are readily decomposed by microorganisms or lost through leaching in the initial period of application [3,43,50]. Alternatively, lower competition among tobacco plants at the early growth stage might have caused large amounts of nutrients (e.g., N or C) to be taken up by microorganisms [27]. However, the priming effect was not observed in grassland soil due to the large amount of native SOC [51]. We did not quantitatively analyze all potential mechanisms of C loss in this study.

### 4.2. Effect of Biochar on the SOC Pool in a Tobacco Field

The uptake of N fertilizer applied by broadcasting depends on subsequent precipitation or irrigation [52]; the fertilizer does not readily move to the root zone under limited precipitation or irrigation, remaining instead in the topsoil [48]. In this study, all of the mineral fertilizers (providing N) were applied by broadcasting near the tobacco plant. Thus, soil TN or MBN contents were higher at 0–15 cm than at 15–30 cm because of the low precipitation (Figure 1) or irrigation in the tobacco field. A similar result was reported by Chen et al. (2016b) [28], who found that 76.5–79.9% of residual urea N (of which 90% was N immobilized in SOM) remained in the topsoil after broadcast application [53]. Furthermore, similar to SOC fractions, soil N also accumulated near the site of biochar placement, as evidenced by the highest soil TN and MBN contents at 0–15 cm under the BFH treatment and at 15–30 cm under the BFB treatment, respectively (Figure 4a,d). Besides influencing water retention, C accumulated in biochar-supplemented soil may stimulate N assimilation and enhance microbial activity [54], thereby increasing soil TN and MBN contents.

In this study, soil $NH_4^+$-N and $NO_3^-$-N contents were significantly higher under the F and BFH treatments than under the BFB and BFBH treatments at 0–15 cm, and higher under the F and BFB treatments than under the BFH and BFBH treatments at 15–30 cm. These results can be explained as follows. Firstly, because there was no biochar

application under the F treatment, less N from the N fertilizers was assimilated and more was directly transformed into $NH_4^+$-N and $NO_3^-$-N. Additionally, biochar decreased inorganic N content via adsorption of N compounds [20,55]. Thus, the soil $NH_4^+$-N and $NO_3^-$-N contents at 15–30 cm under the F treatment were not only higher than those in the same soil layer under the BFB and BFBH treatments, but also higher than BFH and BFBH treatments at 0–15 cm (Figure 4b,c). This suggests that without biochar application, the loss of inorganic N due to leaching is increased. The lower crop yield under no-biochar application compared to biochar application may verify the loss of inorganic N (Figure 5). Secondly, the co-application of biochar and mineral fertilizer can increase N utilization efficiency and stimulate N uptake [16,56] and crop yield (Figure 5); this could also explain the lower $NH_4^+$-N and $NO_3^-$-N contents at 0–15 cm under biochar treatment (except for BFB) than at 15–30 cm without biochar treatment. Thirdly, soil N accumulates in biochar-supplemented soil as a result of water retention and high C:N ratio, which promotes N immobilization or assimilation [11,57]. Furthermore, biochar produced from lignocellulosic feedstock tends to cause net N immobilization over the short term [16], leading to the consumption of $NH_4^+$-N and $NO_3^-$-N used for nitrification and denitrification, respectively [58]. However, N immobilized in the presence of biochar accounts for a negligible proportion of total inorganic N and is released to partially meet crop N demands [15]. Compared to other biochar placement methods, hole placement increased $NH_4^+$-N and $NO_3^-$-N contents in the soil layer at 0–15 cm, while band placement increased the contents at 15–30 cm.

In the layer at 15–30 cm, a large amount of biochar applied in the band (i.e., BFB treatment) might have promoted the movement of broadcast-applied N fertilizer in the topsoil to the biochar-supplemented layer by increasing the C:N ratio [7]. Biochar mixed into the subsoil (10–20 cm) has been shown to enhance the rate of wetting front migration and water infiltration into silty clay soil, thereby stimulating the migration of soil inorganic N from topsoil to subsoil [29]. Biochar is generally reported to improve root growth, including root biomass and root length intensity [9,59,60], which could also promote the movement of N to the biochar-supplemented layer. Indeed, $NH_4^+$-N and $NO_3^-$-N contents at 15–30 cm were higher under BFB treatment than under other biochar treatments, even in a tobacco field with limited precipitation or irrigation. This suggests that band placement of biochar stimulated the soil layer at 15–30 cm to capture more inorganic N. Thus, the biochar placement method can influence inorganic N distribution in tobacco fields over the short term.

Our results also showed that the soil content of inorganic N—especially $NH_4^+$-N—decreased quickly with the tobacco growth period, which may be attributable to net immobilization via microbial assimilation [11]. Biochar combined with inorganic N fertilizers increased nitrification in soils, thereby directly reducing soil $NH_4^+$-N content [16,61,62]. Additionally, as tobacco growth proceeded, there was a greater decrease in $NO_3^-$-N content under the F treatment (from 38.72 to 22.75 mg kg$^{-1}$ at 0–15 cm and from 31.19 to 19.75 mg kg$^{-1}$ at 15–30 cm) than under biochar treatments. This may be explained by the increased nitrification after biochar and inorganic N fertilizers were incorporated into the soil, which increased the $NO_3^-$-N source. Alternatively, N that is immobilized or adsorbed N following short-term biochar amendment will eventually be released through microbial turnover due to physical disruption, predation, or starvation when C-rich substrates are depleted [16]. The decrease in soil MBN content (especially in the layer at 0–15 cm) from the vigorous growth stage to maturity may promote N release via microbial turnover. The higher microbial activity (supported by the increased MBC and MBN contents) under biochar treatments as compared to the control (no-biochar) treatment may also increase N turnover. Biochar application reduced inorganic N ($NH_4^+$-N and $NO_3^-$-N), but this was not necessarily detrimental to crop production (Figure 5). Moreover, the short-term application of biochar increased the inorganic N supply through remobilization of biochar-adsorbed N in the later period of tobacco plant growth. Thus, short-term biochar application could

reduce the N loss in the early growth stage and supply inorganic N (especially for $NO_3^--N$) in the later growth stage of tobacco.

Overall, biochar incorporation can greatly increase soil total C and N pools over the short term, especially in the biochar-supplemented layer, but does not enhance the soil stable C pool because of increased soil LOC and microbial activity. The loss of C after the short-term application of biochar should be quantitatively analyzed in order to determine the C budget. Additionally, biochar application provided inorganic N (especially $NO_3^--N$) at later growth stages, especially in the biochar-supplemented layer. Therefore, the biochar placement method as well as the residual effects of N should be considered in land management practices.

## 5. Conclusions

Biochar application increased soil TOC, TN, and labile C (DOC and ROC) contents, as well as the CPMI and microbial activity (MBC and MBN), which do not favor C sequestration; it also reduced soil $NH_4^+-N$ and $NO_3^--N$ contents in a tobacco field compared to the no-biochar treatment, which therefore reduced the loss risks of inorganic N. These effects were more obvious in the biochar-supplemented soil layer. Biochar application reduced inorganic N ($NH_4^+-N$ and $NO_3^--N$) leaching in the early growth stage of tobacco plants, whereas it supplied inorganic N (mainly $NO_3^--N$) in later growth stages. Finally, the placement methods of biochar in soil influenced the concentrations of soil N fractions; band placement of biochar especially promoted vertical infiltration of $NH_4^+-N$ and $NO_3^--N$. These findings provide insight into how soil physicochemical properties can potentially be improved by biochar application and placement method in continuously cropped fields to maximize crop production. In the future, more works related to nutrient loss through runoff (e.g., surface and subsurface) in biochar-applied soil (different placement methods) are necessary.

**Author Contributions:** R.H.: Conceptualization, Methodology, Writing—draft; B.L.: Conceptualization, Funding acquisition, Reviewing and editing; Y.C.: Data curation, Drawing; Q.T.: Data curation, Visualization, Validation; Q.X.: Visualization, Reviewing and editing; D.W.: Writing—review and editing; X.G.: Writing—review and editing; Q.L.: Reviewing and editing; X.T.: Writing—review and editing; C.W.: Conceptualization, Reviewing and editing, Funding acquisition, Project administration. All authors have read and agreed to the published version of the manuscript.

**Funding:** This research was funded by the National Key Research and Development Program of China (grant number: 2018YFD0800605); The Applied Basic Research Programs of Sichuan Science and Technology Department (grant number: 2018JY0002); the Key Program of China National Tobacco Corporation Sichuan (grant number: SCYC202004); and the Science and Technology Project of Chongqing Municipal Education Commission (grant number: KJZD-K20180410).

**Data Availability Statement:** The data that support the findings of this study are available on request from the corresponding author.

**Acknowledgments:** The authors thank the funding of Chongqing Key Laboratory of Soil Multi-scale Interfacial Process.

**Conflicts of Interest:** The authors declare no conflict of interest.

## Abbreviations

ANOVA, analysis of variance; BFB, biochar band placement + fertilizer; BFBH, biochar band and hole placement + fertilizer; BFH, biochar hole placement + fertilizer; C, carbon; CPI, carbon pool index; CPMI, carbon pool management index; DOC, dissolved organic carbon; F, fertilizer; K, potassium; LI, lability index; LOC, labile organic carbon; MBC, microbial biomass carbon; MBN, microbial biomass nitrogen; N, nitrogen; $NH_4^+-N$, ammonium nitrogen; NLC, nonlabile carbon; $NO_3^--N$, nitrate nitrogen; P, phosphorus; POC, particle organic carbon; ROC, readily oxidizable carbon; SOC, soil organic carbon; TN, total nitrogen; TOC, total organic carbon.

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
