# Peer review of "Biochar Application Increases Labile Carbon and Inorganic Nitrogen Supply in a Continuous Monocropping Soil"

_land, doi:10.3390/land11040473_

Round 1

Reviewer 1 Report

The authors are highly suggested to add the weather data as it will affect the mineralization in the soil which will definitely affect the C and N dynamics in the soil 

No paper from 2022 has been consulted. The authors should consult the recent articles related to the search work. The following papers may be consulted and cited 
https://doi.org/10.1007/s12517-022-09548-8

 https://doi.org/10.1016/j.scitotenv.2021.150444

Author Response

Response to reviewers (land-1612679)

On behalf of my co-authors, I would like to thank anonymous reviewers for dedicating his/her time to provide comments and criticism. The reviewers raise important issues which have helped us to improve our manuscript. We have completed a substantial revision and our changes are highlighted using the “Track Changes” function in the revised manuscript. Our point-by-point response to reviewer’s comments is as follows:

Reviewer # 1

  1. The authors are highly suggested to add the weather data as it will affect the mineralization in the soil which will definitely affect the C and N dynamics in the soil.

Response: Thanks for the reviewer’s comment. The weather data was added in the revised manuscript as below:

Figure 1. Average temperature, average precipitation and sunshine duration in study site

  1. No paper from 2022 has been consulted. The authors should consult the recent articles related to the search work. The following papers may be consulted and cited 
    https://doi.org/10.1007/s12517-022-09548-8
    https://doi.org/10.1016/j.scitotenv.2021.150444

Response: Thanks for the reviewer’s comment and the suggested references were cited in the revised manuscript.

Reviewer 2 Report

I thank the authors of the manuscript entitled “Biochar application increases labile carbon and inorganic nitrogen supply in a continuous monocropping soil” (Huang and co-authors), for significantly improving a previous version that I already reviewed of this interesting study. As said before, I really liked the experimental design, which is complex, in a good way. All other things I mentioned before in a previous revision. On the current one, I have some minor comments below.

Specific comments:

Abstract

L.18, 19. Change “to soil layer” for “to the soil layer”.

L.30. Change “N pool” for “N pools”.

L.31. Delete “the main effects of”.

Introduction

L.39. Change “capacity and N/P contents” for “capacity, and N/P contents”.

L.54. Change “found enhanced denitrification rate in sediment” for “found to enhance the denitrification rate in sediments”.

L.54-55. Is this (promoted N2O emissions) a good thing?

L.64. Change “jeopardizing agricultural sustainability” for “jeopardizing the agricultural sustainability of this crop”.

L.73. Change “layers. Biochar” for “layers. For example, biochar”.

Materials and Methods

L.89-91. Add the word “of” between the soil chemical property and the quantity.

L.94-95. How did you ensure that the pH was exactly 9.65? Clarify this in the text.

L.121. Here, “around” or exactly 2250 plants were planted? By the way, I think that “planted” or “sown” is a better term.

L.135. Change “stages and at plant” for “stages, and at plant”.

L.136. Please explain to the reader how did you come up with this total number of raw samples (600). Ie., 2 soil layers X 4 growth stages X 15 plots X 5 replicates; what I mean is that this previous explanation should be explicit in the manuscript.

L.141. Change “of soil total organic C” for “of total soil organic C”.

L.149. Change “study” for “studies”.

L.165-169. Did you check the normality of the data? You should. Please clarify this in the text.

Results

L.178. Change “was under” for “was found under”.

L.199. Change “was under BFH” for “was found under the BFH”.

L.209. Change “was under” for “was found under”.

L.264. What do you mean by “soil resources”?

L.268. Change “by BFBH” for “by the BFBH”.

L.286. Change “treatment, except for BFH treatment” for “treatments, except for BFH”.

Discussion

L.301. Find a better verb than “supported”.

L.302. Change “level” for “content”.

L.306. Change “by biochar application” for “by its application”.

L.313. Change “to broadcast” for “to a broadcast”.

L.317-320. Is this conjecture a general one or just for the particular conditions of this study? Specify.

L.322. Change “the BFH (or BFB) treatment” for “the BFH or BFB treatments”.

L.330. Change “On the other hand” for “In contrast”.

L.336. Change “microorganism” for “microorganisms”. Change “in impoverished” for “in an impoverished”.

L.337-340. So, temperature should be increased maybe? If so, add that in the manuscript.

L.380. Change “efficiency, stimulate” for “efficiency, and stimulate”.

L.410. Change “under F treatment” for “under the F treatment”.

L.411. Change “biochar treatment” for “biochar treatments”.

L.430. Here, delete the word “for”.

L.432. Change “biochar” for “the biochar”.

Conclusions

L.437. What could be the consequences of reducing the contents of these N forms?

Author Response

Response to reviewers (land-1612679)

On behalf of my co-authors, I would like to thank anonymous reviewers for dedicating his/her time to provide comments and criticism. The reviewers raise important issues which have helped us to improve our manuscript. We have completed a substantial revision and our changes are highlighted using the “Track Changes” function in the revised manuscript. Our point-by-point response to reviewer’s comments is as follows:

Reviewer # 2

I thank the authors of the manuscript entitled “Biochar application increases labile carbon and inorganic nitrogen supply in a continuous monocropping soil” (Huang and co-authors), for significantly improving a previous version that I already reviewed of this interesting study. As said before, I really liked the experimental design, which is complex, in a good way. All other things I mentioned before in a previous revision. On the current one, I have some minor comments below.

  1. L.18, 19. Change “to soil layer” for “to the soil layer”.

Response: Agreed and changed.

  1. L.30. Change “N pool” for “N pools”.

Response: Thanks for the reviewer’s comment here. Revision was proceeded accordingly.

3.L.31. Delete “the main effects of”.

Response: Thanks for the reviewer’s comment here. Revision was proceeded accordingly.

  1. L.39. Change “capacity and N/P contents” for “capacity, and N/P contents”.

Response: Thanks for your comments here and “capacity and N/P contents” was changed to “capacity, and N/P contents”.

  1. L.54. Change “found enhanced denitrification rate in sediment” for “found to enhance the denitrification rate in sediments”.

Response: Thanks for your comments. Revision was proceeded accordingly.

  1. L.54-55. Is this (promoted N2O emissions) a good thing?

Response: We are sorry for the confusion. Increased emission of N2O in agricultural lands is a negative effect for the environment. In order to avoid any confusion, “and promoted N2O emissions” was removed and this sentence was revised as below:

For example, biochar application was found to enhance the denitrification rate [17], increased the N uptake by plants [10], decreased dissolved organic N losses [8,18-19], and caused net N immobilization in soils [16].”

  1. L.64. Change “jeopardizing agricultural sustainability” for “jeopardizing the agricultural sustainability of this crop”.

Response: Agreed and changed.

  1. L.73. Change “layers. Biochar” for “layers. For example, biochar”.

Response: Thanks for your comments. Revision was proceeded accordingly.

  1. L.89-91. Add the word “of” between the soil chemical property and the quantity.

Response: Thanks for your comments. Revision was proceeded accordingly.

  1. L.94-95. How did you ensure that the pH was exactly 9.65? Clarify this in the text.

Response: We are sorry for the confusion. Relevant sentences were revised as below:

Biochar was obtained from Sichuan Meijia Biomass Energy Co. LTD, China, and was prepared by colza straw pyrolysis at a temperature of 400°C–500°C under anaerobic conditions. The N, P, and K contents of biochar were 5.9 g kg1 (total N [TN]), 0.91 g kg1 (total P), and 26.0 g kg1 (total K), respectively; and the C and ash contents were 481.4 and 20.8 g kg1, respectively. The pH of biochar was 9.65 (measured at biochar:water=1:10).

  1. L.121. Here, “around” or exactly 2250 plants were planted? By the way, I think that “planted” or “sown” is a better term.

Response: Thanks for your comments. This sentence was changed to “2250 flue-cured tobacco plants were planted in total” accordingly.

  1. L.135. Change “stages and at plant” for “stages, and at plant”.

Response: Thanks for your comments. Revision was proceeded accordingly.

  1. L.136. Please explain to the reader how did you come up with this total number of raw samples (600). Ie., 2 soil layers X 4 growth stages X 15 plots X 5 replicates; what I mean is that this previous explanation should be explicit in the manuscript.

Response: Thanks for your comments. Soil samples were collected from two layers (0~15 cm and 5~30 cm). Five random samples from each plot. The total number of soil samples is 2 layer × 5 samples per plot × 15 plots × 4 stages = 600. It is noted that five samples were mixed as one pooled sample. Therefore, 120 pooled soil samples were measured. All the information was added in the Section 2.4 as below:

Two soil layers including topsoil (0–15 cm) and subsoil (15–30 cm) were sampled at rosette, vigorous, and squaring growth stages, and at plant maturity (total number of raw samples is 2 layer × 5 samples per plot × 15 plots × 4 stages = 600 ). Five random samples from each plot and each layer were pooled for the analysis of soil LOC fractions (microbial biomass C [MBC] [33], dissolved organic C [DOC] [6-7], readily oxidizable C [ROC] [6-7], and particle organic C [POC] [34]) and soil mineral N fractions (ammonium N [NH4+-N] and nitrate N [NO3-N]) (120 pooled samples were measured).

  1. L.141. Change “of soil total organic C” for “of total soil organic C”.

Response: Thanks for your comments. Revision was proceeded accordingly.

  1. L.149. Change “study” for “studies”.

Response: Thanks for your comments. Changes were made accordingly.

  1. L.165-169. Did you check the normality of the data? You should. Please clarify this in the text.

Response: Thanks for your comments. A sentence of “Normality of datasets was assessed by the Shapiro-Wilk test” was added in the revised manuscript.

  1. L.178. Change “was under” for “was found under”.

Response: Thanks for your comments. Changes were made accordingly.

  1. L.199. Change “was under BFH” for “was found under the BFH”.

Response: Thanks for your comments. Changes were made accordingly.

  1. L.209. Change “was under” for “was found under”.

Response: Thanks for your comments. Changes were made accordingly.

  1. L.264. What do you mean by “soil resources”?

Response: The description of “soil resources” in the original manuscript was not clear. Soil nutrient stoichiometry, such as TOC:TN ratio, can be used to robustly indicate the quality of soil organic matter (Kirkby et al. 2011; Heyburn et al. 2017; Wang et al. 2017) and to reveal nutrient limitations (Huang et al. 2021). This sentence has been changed to “The TOC:TN ratio, which a measure of soil organic quality and nutrient limitations (Kirkby et al. 2011; Huang et al. 2021), was higher…” to make the reader clear.

Kirkby, C.A., Kirkegaard, J.A., Richardson, A.E., Wade, L.J., Blanchard, C., Baten, G., 2011. Stable soil organic matter: a comparison of C/N/P:S ratios in Australian and other world soils. Geoderma 162, 197–208. https://doi.org/10.1016/j. geoderma.2011.04.010.

Heyburn, J., McKenzie, P., Crawley, M.J., Fornara, D.A., 2017. Effects of grassland management on plant C/N/P stoichiometry: implications for soil element cycling and storage. Ecosphere 8 (10), e01963. https://doi.org/10.1002/ecs2.1963.

Wang, N., Fu, F., Wang, B., Wang, R., 2017. Carbon, nitrogen and phosphorus stoichiometry in Pinus tabulaeformis forest ecosystems in warm temperate Shanxi Province, north China. J. Forest Res. 6, 1665–1673. https://doi.org/10.1007/sl1676-017-0571-8.

Huang, R., Lan, T., Song, X., Li, J., Ling, J., Deng, O., Wang, C., Gao, X., Li, Q., Tang, X., Tao, Q., Zeng, M., 2021. Soil labile organic carbon impacts C:N :P stoichiometry in urban park green spaces depending on vegetation types and time after planting. Appl. Soil Ecol. 163, 103926. https://doi.org/10.1016/j.apsoil.2021.103926.

  1. L.268. Change “by BFBH” for “by the BFBH”.

Response: Thanks for your comments. Revision was proceeded accordingly.

  1. L.286. Change “treatment, except for BFH treatment” for “treatments, except for BFH”.

Response: Thanks for your comments. Changes were made accordingly.

  1. L.301. Find a better verb than “supported”.

Response: Thanks for your comments. We have changed “supported” for “evidenced”.

  1. L.302. Change “level” for “content”.

Response: Thanks for your comments. Changes have been made accordingly.

  1. L.306. Change “by biochar application” for “by its application”.

Response: Thanks for your comments. Changes have been made accordingly.

  1. L.313. Change “to broadcast” for “to a broadcast”.

Response: Thanks for your comments. Changes have been made accordingly.

  1. L.317-320. Is this conjecture a general one or just for the particular conditions of this study? Specify.

Response: Thanks for the reviewer’s comment here. This conjecture is just for the particular conditions of this study. This sentence was revised as below:

This suggests that SOC does not readily mobilized along the soil profile but accumulates near the site of biochar placement (ie, the soil layer to which the biochar was added) in this study. The potential reasons are the limited precipitation or irrigation events [28] and short-term application during the tobacco growth period.

  1. L.322. Change “the BFH (or BFB) treatment” for “the BFH or BFB treatments”.

Response: Thanks for your comments. Changes were made accordingly.

  1. L.330. Change “On the other hand” for “In contrast”.

 Response: Thanks for your comments. Changes were made accordingly.

  1. L.336. Change “microorganism” for “microorganisms”. Change “in impoverished” for “in an impoverished”.

Response: Thanks for your comments. Changes were made accordingly.

  1. L.337-340. So, temperature should be increased maybe? If so, add that in the manuscript.

Response: Thanks for the reviewer’s comment here. We are sorry for the confusion, and revisions were conducted as below:

Additionally, biochar used in this study was pyrolyzed at 400°C–500°C with an anaerobic condition, which resulted an incomplete oxidization. Accordingly, ROC fractions released from biochar itself could contribute increases of soil ROC [3]”

  1. L.380. Change “efficiency, stimulate” for “efficiency, and stimulate”.

Response: Thanks for your comments. Changes were made accordingly.

  1. L.410. Change “under F treatment” for “under the F treatment”.

Response: Thanks for your comments. Changes were made accordingly.

  1. L.411. Change “biochar treatment” for “biochar treatments”.

Response: Thanks for your comments. Changes were made accordingly.

  1. L.430. Here, delete the word “for”.

Response: Thanks for your comments. Changes were made accordingly.

  1. L.432. Change “biochar” for “the biochar”.

 Response: Thanks for your comments. Changes were made accordingly.

  1. L.437. What could be the consequences of reducing the contents of these N forms?

Response: Thanks for the reviewer’s comment here. Reducing the contents of inorganic N forms (NH4+-N and NO3-N) in soils is a benefit for biochar application. Because those inorganic N could be lost through leaching in the early growth stage of tobacco plants. Particularly, if those nutrients transported to water body, it may result eutrophication. To make the reader clear, the conclusion was revised as below:

Biochar application increased soil TOC, TN, and labile C (DOC and ROC) contents as well as the CPMI and microbial activity (MBC and MBN), which do not favor C sequestration; and also reduced soil NH4+-N and NO3-N contents in a tobacco field compared to the no-biochar treatment, which therefore reduce the loss risks or inorganic N.”

Round 2

Reviewer 1 Report

The authors have improved the manuscript as per suggestions. The paper may be accepted

Author Response

On behalf of my co-authors, I would like to thank the reviewer for dedicating his/her time to review our work